# Decellularized Human Umbilical Tissue-Derived Hydrogels Promote Proliferation and Chondrogenic Differentiation of Mesenchymal Stem Cells

**DOI:** 10.3390/bioengineering9060239

**Published:** 2022-05-30

**Authors:** Faiza Ramzan, Sobia Ekram, Trivia Frazier, Asmat Salim, Omair Anwar Mohiuddin, Irfan Khan

**Affiliations:** 1Dr. Panjwani Center for Molecular Medicine and Drug Research, International Center for Chemical and Biological Sciences, University of Karachi, Karachi 75270, Pakistan; faizakhanmeo@gmail.com (F.R.); sobiaekram25@gmail.com (S.E.); asmat.salim@iccs.edu (A.S.); omairmohiuddin1@gmail.com (O.A.M.); 2Obatala Sciences Inc., New Orleans, LA 70148, USA; trivia.frazier@obatalasciences.com

**Keywords:** hydrogel, scaffold, cartilage, human umbilical cord tissue, chondrogenic differentiation

## Abstract

Tissue engineering is a promising approach for the repair and regeneration of cartilaginous tissue. Appropriate three-dimensional scaffolding materials that mimic cartilage are ideal for the repair of chondral defects. The emerging decellularized tissue-based scaffolds have the potential to provide essential biochemical signals and structural integrity, which mimics the natural tissue environment and directs cellular fate. Umbilical cord-derived hydrogels function as 3D scaffolding material, which support adherence, proliferation, migration, and differentiation of cells due to their similar biochemical composition to cartilage. Therefore, the present study aimed to establish a protocol for the formulation of a hydrogel from decellularized human umbilical cord (DUC) tissue, and assess its application in the proliferation and differentiation of UC-MSCs along chondrogenic lineage. The results showed that the umbilical cord was efficiently decellularized. Subsequently, DUC hydrogel was prepared, and in vitro chondral differentiation of MSCs seeded on the scaffold was determined. The developed protocol efficiently removed the cellular and nuclear content while retaining the extracellular matrix (ECM). DUC tissue, pre-gel, and hydrogels were evaluated by FTIR spectroscopy, which confirmed the gelation from pre-gel to hydrogel. SEM analysis revealed the fibril morphology and porosity of the DUC hydrogel. Calcein AM and Alamar blue assays confirmed the MSC survival, attachment, and proliferation in the DUC hydrogels. Following seeding of UC-MSCs in the hydrogels, they were cultured in stromal or chondrogenic media for 28 days, and the expression of chondrogenic marker genes including *TGF-β1*, *BMP2*, *SOX-9*, *SIX-1*, *GDF-5*, and *AGGRECAN* was significantly increased (* *p* ≤ 0.05, ** *p* ≤ 0.01, *** *p* ≤ 0.001). Moreover, the hydrogel concentration was found to significantly affect the expression of chondrogenic marker genes. The overall results indicate that the DUC-hydrogel is compatible with MSCs and supports their chondrogenic differentiation in vitro.

## 1. Introduction

Tissue engineering is a promising approach for the repair and regeneration of cartilaginous tissue using a three-dimensional scaffold that mimics the native tissue microenvironment [1]. Cartilage defects are a well-known clinical problem and a leading cause of disability and economic burden worldwide. Approximately 178 million people were reported with skeletal defects in 2019 [2]. Such defects usually occur as a consequence of physical trauma, microtrauma, ischemia, subsequent necrosis or cancer, and possible genetic predisposition in case of nontraumatic osteochondral defects [3]. As a result of these defects, chronic pain usually occurs, which compromises the mobility of joints and diminishes the quality of life [4]. Regenerative medicine brings an advanced approach for regenerating defective tissue that cannot be repaired by normal physiological healing. Several natural and synthetic biomaterials support and maintain tissue regeneration. However, these approaches often fall short in providing an optimal environment for successful tissue regeneration [5]. To improve the biomaterial’s capability to better mimic the natural tissue microenvironment and enhance its therapeutic efficacy, decellularized tissue-derived scaffolds have been proposed with the potential to provide essential biochemical signals and structural support, thus mimicking the natural tissue environment [6,7,8,9]. Each type of tissue has a unique composition of ECM, primarily containing collagen and proteoglycans, which help in the regulation of proliferation, migration, and morphological features of cells based on the provided cues [6,10], which is a key requirement for the enhancement of repair and regeneration of organs and tissues in the body.

Cell-based therapy has attracted global attention in the discipline of regenerative medicine; however, restricted cell propagation, malignant transformation, and senescence limit cell therapy applications for therapeutic purposes [11]. Currently, available treatment options do not provide an optimal solution for osteochondral regeneration. Therefore, further exploration for suitable management of chondral regeneration is required [12]. The study of regenerative medicine and tissue engineering (TE) explores new scaffolds and develops new advances in the repair and regeneration of defective tissue [13]. Decellularized matrices (DM) are prepared using physical, chemical, and biological methods to eliminate cellular components from tissues and organs [14,15,16]. Some of the components that remain after decellularization include extracellular matrix (ECM) proteins including glycosaminoglycans, collagen, proteoglycans, fibronectin, and growth factors, which can potentially support the maintenance of proliferation and differentiation of cells. Several studies have reported that decellularized tissue present decreased risk of inflammation and host immune reactions due to the removal of antigenic cellular components of the tissue [17,18,19,20]. Scaffolds based on metals and synthetic and natural polymers have not proven to be satisfactory for tissue regeneration and present certain disadvantages, including being allergenic to some patients and having low mechanical load-bearing properties, which make them unreliable for in vivo implantation. ECM and ECM-mimicking scaffolds enable the constructive remodeling of different tissues in the body and are better suited for clinical applications [21]. Therefore, incorporating DM scaffolds aids in a favorable way of developing a setting that better imitates the native tissue microenvironment for injured tissue [21]. DMs are being tested widely for the engineering and regeneration of functional tissue, including cartilage, bone, amniotic membrane, skeletal muscle, and dental pulp [5]. In addition, injectable biomaterials are considered advantageous due to their minimally invasive mode of application to the lesion, which prevents tissue damage during implantation [22,23,24]. Umbilical cord (UC) obtained from the perinatal tissue is routinely discarded as medical waste and is therefore a relatively easily accessible human tissue [25]. UC-derived scaffolds loaded with MSCs have proven to be promising biomaterial composites for the treatment of chondrogenic defects [26]. Wharton’s jelly ECM is rich in proteoglycans, hyaluronic acid, elastin fiber, collagen, and growth factors, i.e., the cluster of differentiation 44 (CD-44)**,** insulin-like growth factor 1 (IGF-1), and platelet-derived growth factor (PDGF) required for proliferation, matrix synthesis, and effective tissue remodeling [25,27,28]. Collagen is the main component of Wharton’s jelly, constituting 50% of the tissue by weight. Hyaluronic acid is the most abundant component of Wharton’s jelly glycosaminoglycan (GAGs). Hydrogels derived from umbilical cord tissue have been shown to regenerate cartilage and other tissue [29,30]. Previous reports have confirmed that UC can be a useful source of decellularized tissue for tissue engineering applications [1,31]. 

The present study is novel and promising in terms of the development of UC tissue-derived hydrogel scaffold for chondrogenic differentiation in vitro. The goal of this research was to develop an UC-ECM-derived scaffold that provides a niche for cartilage development and regeneration. In this study, we first characterized decellularized human UC scaffold and then tested its capability to support chondrogenesis of MSCs. 

## 2. Materials and Methods

### 2.1. Ethical Approval and Umbilical Cord Collection

Fresh human umbilical cord tissue samples (*n* = 15; age = 26 ± 7.8) were collected after written informed consent from donors undergoing elective cesarean section. The study had been approved and reviewed by an independent ethics committee (IEC-009- UCB-2015) of the Dr. Panjwani Center for Molecular Medicine and Drug Research, International Center for Chemical and Biological Sciences, University of Karachi. 

### 2.2. Preparation of Decellularized Human Umbilical Cord (DUC) Tissue

Umbilical cord tissues were collected, pooled together, washed with deionized (DI) water, and sliced into 6–7 cm pieces before decellularization. The tissue was frozen/thawed 3 times, followed by a 24 h wash in DI water. Subsequently, the tissue was subjected to osmotic shock, alternating with 1 M NaCl hypertonic solution for 4 h, followed by an overnight wash in DI water. Thereafter, the tissue was incubated with 0.05% trypsin for 2 h at 37 °C, and washed with DI water for 4 h. The tissue was then subjected to non-ionic detergent 1% Triton-X-100 for 24 h under constant agitation. The obtained tissue was washed overnight with DI water to remove the detergent, followed by incubation in 1 M NaCl solution for 4 h, and washing with DI water for 24 h. Finally, the decellularized umbilical cord (DUC) tissue was stored at −20 °C.

### 2.3. Confirmation of Decellularization 

#### 2.3.1. DNA Content Determination

DNA from native and decellularized tissues was extracted using a genomic DNA purification kit (Thermo scientific #K0721) according to the manufacturer guidelines. Fifty milligrams of tissue was physically disrupted by freezing and grinding with a mortar and pestle under liquid nitrogen and homogenized with a tissue homogenizer (Ultra-Turrax, IKA-Werke). The tissue was then digested with proteinase K for 2 h at 56 °C, followed by adding lysis solution. Then the lysate was transferred to the Gene JET Genomic DNA purification column and centrifuged to elute the DNA. The DNA was eluted and per mg concentration of tissue was determined at 260 nm wavelength by UV–visible spectrophotometer (Nanodrop 2000, Thermo Fisher Scientific, Wilmington, NC, USA).

#### 2.3.2. Agarose Gel Electrophoresis

Agarose gel electrophoresis was performed to further validate the DNA elimination from DUC tissue. One percent agarose gel (Sigma, Saint Louis, MO, USA) was transferred into electrophoresis apparatus (Sub-Cell GT Agarose Electrophoresis Systems, Bio-Rad, Hercules, CA, USA), submerged in 1X TBE buffer. Samples and ladder were loaded into the wells, run at 70 V for 1 h, and documented in the visualization system (AlphaEAse FC imaging system, FluorChemTM, Alpha Innotech, San Leandro, CA, USA). The intensity of the DNA bands was analyzed.

### 2.4. Histological Examination 

Histological examination was performed to inspect the efficacy of decellularization. DAPI (4′,6-diamidino-2-phenylindole) and hematoxylin and eosin (H and E) staining were used to visualize the nuclear content, while Alcian blue and Masson’s trichrome staining were used to check ECM preservation before and after decellularization. 

#### 2.4.1. Fixation, Embedding, and Cryosectioning of Harvested Tissue

Tissue processing for histological analysis was performed as previously described [32,33]. Briefly, native and DUC tissue were fixed with 4% paraformaldehyde at room temperature. The tissue sections were then embedded in the optimal cutting temperature (OCT) medium (Surgipath, FSC22, Leica Microsystems, Wetzlar, Germany), and frozen at −20 °C. Tissue blocks were place on Cryotome E (Shandon, Thermo Electron Corporation, Horsham, UK), and 7–10 µm thick sections were cut and transferred to charged gelatin-coated slides (2105, SuperFrost, Carl Roth, Roth, Germany). The slides were stored at −20 °C. 

#### 2.4.2. DAPI Staining

Nuclei of native and DUC samples were stained with 1 µg/mL of DAPI in PBS for 10 min at room temperature (RT). The slides were washed with PBS, mounted with an aqueous mounting medium, and coverslips were gently placed over them. The stained slides were evaluated under fluorescent microscope (NiE, Nikon, Tokyo, Japan). Images were acquired with CCD camera, and processed by imaging software NIS-Elements (Ti2, Nikon, Tokyo, Japan).

#### 2.4.3. Hematoxylin and Eosin Staining

The nuclear and cytoplasmic content were stained with H and E. Hematoxylin was added to the slides for 5 min, then washed with DI water carefully to wipe out excess stain. Then, the sections were counterstained with eosin Y for approximately 30 s to stain the cytoplasm, followed by washing. Once the sections were properly stained, the slides were subjected to 70%, 90%, and 100% isopropanol washes followed by xylene wash for 5 min each to dehydrate the specimens. Each slide was then mounted by DPX mounting medium and observed under microscope (NiE, Nikon, Tokyo, Japan).

#### 2.4.4. Quantification of GAG

Native and DUC tissue and hydrogel with the mean weight of 50 mg sample in PBS were digested with 300 μg/mL papain (Sigma, Saint Louis, MO, USA) at 58 °C overnight, and the digested samples were placed in 1,9-dimethyl-methylene blue (DMMB) solution. The absorbance was measured at wavelengths 540 nm and 595 nm. The sulfated GAG (sGAG) content μg per mg dry weight of tissue was calculated using chondroitin sulfate standard curve (Sigma, USA). 

#### 2.4.5. Alcian Blue Staining 

To qualitatively analyze the GAG content of ECM in the native and DUC tissue, 1% Alcian solution in 3% glacial acetic acid was used to stain the sulfated GAGs. The sections were washed with DI water, followed by incubation with Alcian working solution for 5 min, the excessive stain was removed by washing with tap water for 30 s, dehydrated with graded alcohol, mounted, and the images were captured under a bright field microscope (NiE, Nikon, Tokyo, Japan).

#### 2.4.6. Masson’s Trichrome Staining

Collagen content of native and DUC tissues was determined by Masson’s trichrome staining kit (Medlines, Manchester, UK) according to provided instructions. Briefly, the tissue sections (5–7 μm thickness) were allowed to set at RT, gently rinsed with DI water, and incubated with stain. Subsequently, sections were washed, dehydrated, and coverslipped. The stained slides were observed under bright field microscope (NiE, Nikon, Tokyo, Japan).

### 2.5. 3D Hydrogel Preparation from Decellularized Umbilical Cord Tissue 

The DUC tissues were frozen at −80 °C overnight and were lyophilized in a freeze-drier (TrioScience, Istanbol, Turkey) for 24 h. The lyophilized DUC samples were then cryo milled to powder, and gel was prepared as previously described [34]. Briefly, the dried DUC powder from 4–5 donors was pooled and digested with 1 mg pepsin/mL (Sigma, USA; then, it was dissolved in 0.05 M HCl) per 10 mg DUC tissue under constant agitation for 48 h at room temperature. Ten and twenty milligram powdered tissue was exposed to UV light for 30 min inside the biosafety cabinet for disinfection, followed by the addition of 1 mL of pepsin/HCL (to make 10 mg/mL and 20 mg/mL hydrogel) in each of the vials for up to 48 h at room temperature with constant agitation to ensure complete digestion and formation of a pre-gel liquid which was then neutralized by the addition of ice-cold solution of 1 M NaOH (1/10th of original digest volume) and 10X PBS (1/10 of final neutralized volume) to attain a pH of 7.4. Neutralized pre-gel solution was allowed to gel by incubation at 37 °C (NU5500E, NuAire, USA) for 20–30 min to be completely turned into hydrogel. Gelation was confirmed either macroscopically or microscopically under phase contrast microscope (TE 2000S Eclipse, Nikon, Tokyo, Japan). 

### 2.6. Hydrogel Characterization

#### 2.6.1. Material Composition by FTIR (Fourier-Transform Infrared) Spectroscopy

Infrared emission spectra of the lyophilized tissue, pre-gel, and hydrogels were acquired in the range of 400–4000 cm^1^ with a resolution of 4 cm^−1^ using (FTIR-1 Bruker Vector 22 spectrometer, Billerica, MA, USA). The spectra of the materials were measured at room temperature, and the data were analyzed using OriginPro 2017 software (Version 2019b, Build No. 9.650169, MA, USA).

#### 2.6.2. Scanning Electron Microscopy (SEM)

A scanning electron microscope (SEM) (Jeol Japan) at a 15 kV accelerating voltage under vacuum was used to observe the morphology of the hydrogel. The specimens were prepared by immersing them in a 4% PFA solution at room temperature for 5 min to preserve the internal structure, and subsequently rinsed 3 times with PBS. The materials were then dehydrated for 10 min using gradient isopropanol (70–100%) washes, and lyophilized. Specimens were then sputter-coated with gold for 30 s (Auto Fine Coater: JEC-3000FC), dried at room temperature, and imaged. SEM equipped software was used to analyze the photographs (SEM Control user interface version 7.11C). To evaluate the porosity of hydrogel, the pore size of each hydrogel was reported as an average size of 30 random pores measured using Image J software (NIH, Besthesda, MD, USA).

#### 2.6.3. Swelling Behavior

The swelling of hydrogels was evaluated gravimetrically by calculating their water absorption capability. The initial weights of hydrogels were first determined before being submerged in phosphate-buffered saline (PBS) solutions of pH 7.4 (10 mL) at 25 °C for 12 h, 24 h, 48 h, 56 h, 72 h, and 96 h. After immersing the samples for a predefined time, they were removed from PBS and weighed. The following equation was used to calculate the amount of water absorbed by hydrogels:Swelling percentage (%) = [(Ws − Wi)/Wi) × 100
where Ws and Wi are the swelling and initial weight of hydrogel, respectively.

### 2.7. Isolation and Characterization of Umbilical Cord-Derived MSCs 

MSCs from human umbilical cord tissue was isolated and characterized as previously described [33,35]. Passage 2–5 MSCs were used for the experiments. 

### 2.8. MSCs Cultured in Decellularized Umbilical Cord (DUC) Hydrogel 

MSCs were resuspended in pre-gel at a concentration of 0.2 million cells per 200 µL and pipetted as domes in 48-well plates. The pre-gels were incubated at 37 °C for 30 min to form hydrogels. The resulting DUC hydrogel scaffolds were dome-shaped with uniform heights. The morphology was analyzed under phase contrast microscope. MSC-seeded hydrogels were supplemented with stromal and chondrogenic media comprising 100 nM dexamethasone, 20 ng TGF-β, 10 ng insulin, and 1000 µM ascorbic acid for up to 28 days for the gene expression analysis. 

#### F-Actin Staining

MSC-seeded DUC hydrogels were fixed by immersion in 4% paraformaldehyde overnight at room temperature, then rinsed with 1X PBS to remove the fixative. Cells were permeabilized with 0.1% Triton-X-100, blocked by incubating with blocking solution (PBS containing 2% BSA and 0.1% Tween 20) at 37 °C for 1 h. The cellular cytoskeleton was stained with Alexa fluor488 phalloidin at 37 °C for 1 h. The nuclei were stained with DAPI (0.5 µg/mL) for 15 min at room temperature. The cells were then washed 5 times with PBS and observed under a fluorescent microscope (Ti2, Nikon, Japan).

### 2.9. Assessment of Cell Viability and Proliferation in Hydrogel 

#### 2.9.1. Live Cell Imaging in 3D Scaffold 

Live cell imaging of 3D cultured MSCs was performed by Calcein AM staining (Invitrogen, Thermo Fisher Scientific, Wilmington, NC, USA). The dye changes its color from nonfluorescent to green fluorescent when hydrolyzed by viable cells. Cell-seeded DUC hydrogels were plated in a 24-well plate, washed with PBS, and incubated with 10 µM working solution of Calcein AM stain for 30 min at 37 °C. The cell viability was examined at respective time points, i.e., 1st, 2nd, 3rd, 5th, and 7th day. The green fluorescent cells were observed under microscope at a wavelength of 469/525 nm and images were captured via CCD camera (Ti2, Nikon, Japan).

#### 2.9.2. Alamar Blue Assay 

The proliferation of MSCs in DUC hydrogel was determined by Alamar blue assay. Briefly, the cells were plated at a density of 1200 cells per 100 µL pre-hydrogel and were incubated at 37 °C to completely turn into hydrogel. The viability and proliferation were measured after 1 day and 1 week of 3D cell culture. Cells were incubated with 0.002% Alamar blue working solution in basal media, the plate was transferred to 37 °C for 30 min and the intensity of reduced Alamar blue with an excitation wavelength at 530–560 nm and emission wavelength at 590 nm were measured by fluorescence microplate reader (NT-003/EQ-V.L.M). Two-dimensional (2D) cultured cells were used as control, and acellular hydrogel was used as negative control. 

### 2.10. Gene Expression Analysis

UC-MSCs were cultured at a density of 0.2 million cells in 200 µL hydrogel, supplemented with chondrogenic induction media. MSC-seeded hydrogels cultured in stromal media and 2D-cultured MSCs were used as controls. All samples were harvested after 28 days of culture. Total RNA was isolated by Trizol (15596026, Invitrogen) extraction method as per manufacturer’s instructions. RNA pellets were collected and dissolved in 25 μL nuclease free water. RNA concentration was measured by absorbance measurement at 260 nm wavelength, 260/280 nm ratio was used to analyze the purity of RNA samples using a UV–visible spectrophotometer (UV-1700, Shimadzu, Japan). Then, cDNA was prepared from 2 μg of RNA using a cDNA synthesis kit (™ RevertAid™, K1622, Thermo Scientific, USA) according to the manufacturer’s instructions. The primers for specific genes were designed by primer blast (https://www.ncbi.nlm.nih.gov/tools/primer-blast/ accessed on 15 May 2020). A detailed description of primer sequences and annealing temperatures is given in Table 1. For gene expression analysis, 10 ng cDNA was mixed with bright Green 2X qPCR Master Mix (Abm, Inc., Vancouver, Canada) using Real-Time PCR System (Cfx96, Biorad, Hercules, CA, USA) for quantitative expression of chondrogenic marker genes. The Ct values of treated and untreated samples were normalized to reference *GAPDH* gene and relative gene expression was determined by 2^−ΔΔCt^ (Livak) method. The experimental groups are shown in Table 2.

### 2.11. Statistical Analysis

Acquired data are presented as mean ± standard deviation and were examined with the help of Student’s *t*-test, one-way analysis of variance (1-way ANOVA) with Bonferroni post-test. A threshold of *p* ≤ 0.05 determined statistical significance (IBM, SPSS, Statistics software version 21) was used to perform statistical analyses. All the measurements were run in triplicate unless otherwise stated.

## 3. Results

### 3.1. Decellularization of Human Umbilical Cord Tissue 

The biological, chemical, and physical methods were used for the decellularization of UC, comprising freeze-thaw, application of osmotic pressure, detergent treatment such as Triton-X-100, and trypsin-based proteolytic digestion (Figure 1A,B). The protocol showed high effectiveness in decellularization and a significant reduction of DNA content compared to native tissue (Figure 1C). 

#### DNA Quantification and Gel Electrophoresis 

The efficiency of the decellularization protocol was confirmed by the evaluation of the DNA content after decellularization. We found that the protocol efficiently removed nucleic acid and proteins from the umbilical cord, and the tissue turned white and transparent as shown in Figure 1B. The DNA content decreased from 174.15 ng ± 2.45 to 12.63 ng ± 2.42 of DNA/mg dry weight of tissue after Triton-X-100 based decellularization. Results from gel electrophoresis showed a recognizable dark, dense high-molecular-weight band of DNA in the native tissue, whereas DNA isolated from decellularized samples show less intense bands, which confirms the elimination of DNA in DUC samples (Figure 1C). 

### 3.2. Biochemical Composition of Native and DUC Tissue 

Hematoxylin and eosin (H and E) and DAPI staining was performed to quantify the effectiveness of the decellularization procedure. Histologically, DUC tissue was porous and devoid of intact cells, while the nuclei or other cellular components were lost during the decellularization process. 

#### 3.2.1. DAPI Staining

The absence of nuclear content was further confirmed qualitatively by DAPI staining in native and DUC samples (Figure 2A,B). The data presented no visible nuclear staining of DUC as compared to native tissue after DAPI staining, indicating the effective removal of DNA from the DUC sample. In contrast, the native sample showed heavy nuclear staining throughout the tissue. 

#### 3.2.2. Hematoxylin and Eosin Staining

The tissue morphology of native and decellularized tissues was further analyzed by H and E staining. The native tissue showed dense eosin stained cytoplasm, and dark purple-stained nuclei; the cell nuclei were found rounded in appearance embedded within cytoplasm (Figure 2A). In contrast, DUC sample showed Eosin-stained ECM, with a lack of cell and nuclear fragments, which confirmed efficient cell removal after the decellularization of samples (Figure 2B).

### 3.3. ECM Structure and Composition of Decellularized Tissue 

#### 3.3.1. sGAG Quantification

GAG analysis indicated that decellularized tissue contained 0.291 ± 0.007 μg/mg while, 10 mg/mL hydrogel contained 3.12 ± 0.13 μg/mg and 20 mg/mL hydrogel contained 3.02 ± 0.19 μg/mg sGAGs compared to native umbilical cord tissue (mean = 0.42 ± 0.005 μg/mg) (Figure 3A).

#### 3.3.2. Glycosaminoglycans (GAGs) Content of Extracellular Matrix

Alcian blue stain showed sulfated glycosaminoglycans (GAGs) in the ECM of tissue, producing an intense blue color around the cell matrix. The results were found to be positive for native and decellularized tissue but the intensity was slightly different in DUC, potentially due to the denaturation of GAGs and reduction in collagen fiber size during the decellularization process. The image indicates the presence of GAGs in native and DUC tissues. The GAGs were also visible in DUC tissue but the staining was less intense in comparison to the native sample (Figure 3B).

#### 3.3.3. Masson’s Trichrome Staining for Collagen 

Masson’s trichrome staining of the human umbilical cord native and decellularized tissues showed collagen-rich ECM in blue and nuclei in dark blue/black distributed in the matrix and cytoplasm. Since the cells were lost during the decellularization process, the DUC matrix obtained was rich in blue stain, representing collagen. Thus, the decellularization process resulted in a porous matrix with preserved collagen fibers and low structural distortion (Figure 3C).

### 3.4. Hydrogel Structural and Mechanical Properties

Hydrogels were successfully prepared from DUC at concentrations of 10 mg/mL and 20 mg/mL. The optimal digestion time was determined based on the degree of ECM digestion. The gel solution remained liquid at 4 °C and turned into hydrogel at 37 °C for 20–30 min, which was observed macroscopically. The procedure for hydrogel preparation is summarized in Figure 4A.

#### 3.4.1. Biochemical Analysis of Hydrogel by FTIR

The cross linking was characterized by comparative analysis of FTIR spectra of the lyophilized sample. Figure 4B shows the FTIR spectra of DUC 10 mg/mL and 20 mg/mL pre-gel and hydrogel; peaks were centered at 3453 cm^−1^ amide A, 2921 cm^−1^ amide B, amide I, II, III was found at wave number of 1600–1301 cm^−1^, and 1106 cm^−1^ show the presence of GAGs. Sharp peaks representing amide I and amide II in both 10 mg/mL and 20 mg/mL hydrogels represented the β-sheet structure which increased during the gelation process compared to 10 mg and 20 mg pre-gel. The absorption bands for hydrogel at amide A band region was broadened due to a large number of hydroxyl groups and the increase in intensity of absorption shows the successful gelation of hydrogel.

#### 3.4.2. Scanning Electron Microscopy

SEM was utilized to examine the fiber surface topography of the fabricated hydrogel. The porous hydrogel scaffolds displayed connected smooth filament topology. The mean pore sizes for 10 mg/mL and 20 mg/mL hydrogels were estimated as 0.89 ± 0.02 and 0.73 ± 0.09 µm, respectively. In comparison to 10 mg/mL hydrogel, 20 mg/mL hydrogel displayed fewer apertures, increased roughness, and a denser fibril network, as seen in Figure 4C. 

#### 3.4.3. Swelling Behavior 

Swelling behavior was tested in PBS (pH 7.4) at 25 °C for 96 h Figure 4D. Swelling was observed in both 10 mg/mL and 20 mg/mL hydrogels due to water imbibition up to 48 h and plateaued thereafter. The results indicate that the porosity and GAGs content of hydrogels allows them to maintain water and gaseous and nutrient uptake, which is essential for cell proliferation.

### 3.5. UC-MSCs Encapsulated in 3D Hydrogel

MSCs were encapsulated into the pre-gel solution and incubated for gelation, and observed under phase contrast microscope, showing spindle-shaped cells with a regular distribution throughout the gel with time as shown in Figure 5A.

#### F-Actin Staining

Morphology of cell aggregates in hydrogel visualized by DAPI and F-actin immunostaining showed the cytoskeleton structure of cells embedded in hydrogel. The figures depict the spindle morphology of MSCs spread out from the edges of the hydrogel, which also confirms the biocompatibility of cells with the hydrogels (Figure 5B). 

### 3.6. Proliferation and Viability in 3D Culture MSCs

#### 3.6.1. Calcein AM Staining

The viability of WJ-MSCs in DUC hydrogels was qualitatively analyzed by Calcein AM stain. The cells were seeded in the hydrogels and cell viability was analyzed at respective time points, i.e., 1st, 2nd, 3rd, 5th, and 7th day, and the adherence and penetration into the hydrogel matrix with time were assessed. The results showed a progressive increase in cell number under fluorescent microscopy. The cells were spindle shape in appearance and were continuously proliferating over time from day 1 to 7, confirming that the hydrogel cell matrix interaction was nontoxic and provided framework support for cell proliferation (Figure 6A).

#### 3.6.2. Alamar Blue Assay

MSC proliferation in 3D culture was assessed by Alamar blue assay. Fluorescence intensity of reduced Alamar blue was measured on day 1 and day 7 in 3D culture, which was normalized to the fluorescence intensity of 2D culture to determine the fold increase in cell viability, and presented as a bar graph. The figure shows increase in the cell number after a week of culture, significant difference in cell growth, and higher cell proliferation at both concentrations of hydrogel groups compared to day 1 in 3D culture. However, the comparison among the gels showed that 10 mg/mL concentration of hydrogel was found to be slightly less proliferative than 20 mg/mL (Figure 6B).

### 3.7. In Vitro Chondrogenic Gene Expression Dynamic

Chondrogenic differentiation potential of MSC-seeded hydrogel was analyzed after 28 days. The relative changes in chondrogenic gene expression profile among 2D-cultured MSCs, 3D-cultured MSCs in stromal media, and chondro-inductive media hydrogel were examined. The experimental groups are mentioned in Table 2. In Figure 7A, the results show significant upregulation of *TGF-β1* and *BMP2* gene expression with *** *p* ≤ 0.001 in 10 mg/mL chondro-inductive hydrogel while *SOX-9* showed significant increase in expression with *** *p* ≤ 0.001 in 20 mg/mL chondro-inductive hydrogel as compared to 2D-cultured MSCs. However, no significant change was noted at day 28 in the *SIX-1, GDF5,* and *AGGRECAN* expression in 10 mg/mL or 20 mg/mL hydrogels as compared to 2D-cultured MSCs. The intercomparison of the two hydrogels groups showed significant differences. *TGF-β1* was significantly increased in the 10 mg/mL hydrogel group, while the 20 mg/mL group showed significantly increased *BMP2* and *SOX-9* expression compared to 10 mg/mL, while *SIX-1*, *GDF5*, and *AGGRECAN* did not display significantly different expression between both concentrations of chondro-inductive media hydrogel. Figure 7B shows significantly increased expression of *BMP2*, *SOX-9*, *SIX-1*, *GDF-5*, and *AGGRECAN* with *** *p* ≤ 0.001 in 20 mg/mL stromal media-cultured hydrogel. However, significant increase in expression of *SOX-9*, *GDF-5*, and *AGGRECAN* with *** *p* ≤ 0.001 in 10 mg/mL stromal media-cultured hydrogel compared to 2D-cultured MSCs was observed. The comparison between 10 mg/mL and 20 mg/mL stromal media hydrogel showed significant differences. *BMP2*, *SOX-9,* and *SIX-1* was upregulated in 20 mg/mL with *** *p* ≤ 0.001 as compared to 10 mg/mL stromal media hydrogel. *GDF5* and *AGGRECAN* showed higher expression in the 10 mg/mL group compared to the 20 mg/mL group, however, *TGF-β1* expression was found to be non-significantly different between both concentrations of stromal media hydrogel. Figure 7C shows significant upregulation of *TGF-β1*, *BMP2*, and *SIX-1* gene expression with ** *p* ≤ 0.01 in 10 mg/mL chondro-inductive hydrogel, while 20 mg/mL chondro-inductive hydrogels displayed significant upregulation of *TGF-β1*, *SOX-9*, and *SIX-1.*


## 4. Discussion

Chondral lesions are a global health challenge; they cause inflammation and loss of bone and cartilage structure [36,37]. Moreover, cartilage has limited self-renewal capability. It has been reported that stem cell-based tissue engineering approaches have the potential to serve as an effective treatment for chondral defects [36]. A number of 3D scaffold-based tissue engineering approaches have been reported for in vivo regeneration of damaged tissues. A diverse range of naturally derived scaffolds are being developed for tissue engineering (TE) purposes. Synthetic materials such as polystyrene poly-l-lactic acid (PLLA) have certain limitations, including the possibility of failure owing to diminished bioactivity and the possibility of cell and tissue apoptosis due to hydrolysis. Natural polymers including collagen, elastin, proteases, and chitosan have processing, purity, and protein denaturation issues. Therefore, there is a need to develop an optimum scaffolding material which is minimally immunogenic, biocompatible, and possesses adequate mechanical strength. For TE applications, decellularized ECM serves as a bioactive scaffold, which is nontoxic, biodegradable, and facilitates cellular adhesion and growth [38]. 3D scaffolds offer a platform to the cells for homing and proliferation by providing a bioactive microenvironment that can regulate cellular fate and direct the synthesis of ECM. Ideally, a 3D scaffold should replicate the natural microenvironment of the cells to achieve a desirable cell phenotype. It has been a standard practice to use autologous or allogeneic tissue for osteochondral regeneration in vivo [26]. The use of biologically active scaffolds prepared from polymers or a decellularized matrix has been extensively investigated for chondrogenic differentiation and efficient reconstruction [39]. Synthetically designed scaffolds do not provide tissue homeostasis in comparison with tissue-derived scaffolds (TDS), which are comprised of native ECM. TDS replicates the native tissue microenvironment, 3D structure, presence of ligands, and bioactive molecules. These scaffolds promote regeneration via stimulation of angiogenesis, infiltration of endogenous cells, and modulation of the immune response [40]. The unique features of TDS ensure the functional remodeling of the lesion area, thus minimizing scar formation [41]. Human UC can serve as a useful biomaterial for the fabrication of scaffolds for TE application [1,31,38,42]. UC contains essential ECM components, hyaluronic acid, and GAGs, which are non-antigenic and support chondrogenic differentiation in vitro [30]. The UC tissue also mimics the cartilaginous tissue by its relatively low avascularity and aneurality [43]. In addition, it has been reported that UC tissue contains collagens, and other ECM proteins [44], which support cell proliferation, homing, and differentiation [1,44]. Hyaluronic acid (HA) in UC tissue makes it highly hydrated, viscous, and suitable for directing stem cells to differentiate along chondrogenic lineage [44,45]. UC tissue ECM contains growth factors and cytokines, which make it useful for tissue engineering applications [44,46]. In recent years, various methods for decellularizing the UC have been designed and tested to generate a scaffold that can sustain cell survival and proliferation while maintaining the native ECM structure and secretion profile [30].

The present study explored a novel method for the preparation of a novel UC tissue-derived hydrogel as an in vitro scaffold which can recapitulate the key physicochemical characteristics of native human tissue. Moreover, the method was designed to minimize ECM damage. The objective of the present study was to decellularize human UC tissue and develop a hydrogel.

The results obtained from the present study showed that DUC hydrogel allowed MSC proliferation and differentiation into chondrogenic lineage in vitro. Previous studies have reported various approaches for the decellularization of tissues with minimal antigenic properties [45]. These studies have reported that sodium dodecyl sulfate (SDS) and Triton-X-100 detergent treatment can cause damage to growth factors and ECM proteins during the decellularization process [46]. Decellularized material can be used as functional 3D scaffolds, promoting cellular adherence and proliferation. It is reported that decellularized urinary bladder, dermis, myocardium, and adipose tissues provide ECM and growth factors, which lead to the recruitment of cells to the lesioned area, resulting in the repair and regeneration of tissues and organs [22,28,47,48,49].

The current study used mild decellularization procedures. The efficiency, quality, and composition of the decellularized tissue were determined by DNA removal and the retention of sGAG and collagen fibrils after the decellularization process. We found that 1% Triton-X-100 treated UC tissue was effective for solubilization of cell membranes and nuclear components and it showed low levels of ECM damage, determined by DNA quantification, and histological analysis by Alcian blue and Masson’s trichrome staining. The quantification of GAG in DUC tissue and hydrogel was also evaluated, which confirmed the maintenance of ECM constituents, thus preserving physiochemical characteristics of native UC tissue. Decellularization protocol must preserve the bioactive molecules to develop a scaffold that can efficiently contribute to regeneration [50]. Our findings are in agreement with previous studies which showed the preservation of ECM after decellularization [44]. The DUC was found to contain sulfated GAGs, which are reported to enhance chondrogenesis, osteogenic differentiation, and improved cell matrix interaction [51]. Another study reported that the presence of chondroitin sulfate enhances the effectiveness of collagen-I based scaffolds in the differentiation of MSCs to chondrocytes [52].

The mechanical properties of the 3D scaffolds also have an important role in supporting and stimulating stem cell proliferation and differentiation. The matrix stiffness can instruct the cell to differentiate into multiple lineages [53]. Softer substrates lead MSCs toward adipogenic lineage and stiffer hydrogel leads MSCs to differentiate into osteocytes and chondrocytes [54,55,56,57,58]. A hybrid hydrogel composed of decellularized UC and silk fibroin hydrogel has displayed good mechanical properties and cell compatibility in cartilage regeneration [1]. FTIR analysis showed the biochemical composition and degree of crosslinking from lyophilized tissue to hydrogel formation; the typical peak of protein secondary structure includes amide A, B, and I, II, and III bands, the absorbance associated with C=O stretching are denoted as amide I, those associated with N—H bending are amide II and amide III, and NH bending associated with (NH stretch coupled with hydrogen bonding) are denoted as amide A, while amide B related to asymmetric stretch of CH2 stretching vibration verified the protein nature of the samples [1,59]. These amide bands are involved in hydrogen bonding to form protein secondary structures and specifically correlate with beta sheets, α-like helices, β-turns, triple helix, side chains, coils, and unordered structures [60]. As a consequence of the gelation process, the molecular chains could become closer and intense peaks were observed, possibly involved in the stabilization of the β-sheet supramolecular structure within hydrogel. In both 10 mg/mL and 20 mg/mL hydrogels, the β-sheet structure was increased after the gelation compared to their respective pre-gel state. The β-sheet structure plays an important role in the hydrogel network formation [59].

Hydrogels are hydrophilic polymer networks that are capable of imbibing large amounts of water. Natural polymers can form hydrogels. Studies showed that the addition of water-soluble polymers improved mechanical properties in terms of gel content, swelling behavior, and gel strength. When a hydrogel is immersed in water, it swells until the osmotic forces that help to extend the polymer network are balanced by the elastic forces from the stretched segments of the polymer. Previous studies [61] showed that the natural hydrogel has porous morphology, and water imbibition capacity may be due to hydrophilic molecules present in DUC tissue-derived hydrogel such as the presence of GAGs, which is consistent with our results, which revealed that 20 mg/mL hydrogel content has a denser and tighter structure with smaller pore size while 10 mg/mL hydrogel has larger pores as analyzed by SEM, with diameters ranging from 0.7–0.8 µm. The porous three-dimensional structure of these hydrogels is suitable for chondrogenic and osteogenic differentiation, as previously reported [1,62]. These scaffolds provide sufficient area for cell growth and migration. Hydrogels used in bone and cartilage regeneration should be made of biocompatible materials and have adequate stability to allow cell activity. Biodegradation of hydrogels is essential for the removal of non-native contents from the regenerated tissue and microstructural remodeling. Regardless of the natural or synthetic polymers employed in their production, the breakdown of hydrogels is primarily influenced by temperature, pH, UV irradiation, ultrasound, and enzymes, among other factors [23]. Previous study reported that the degradation of ECM proteins in decellularized adipose tissue (DAT) hydrogel by MMPs allows the cells to remodel the surrounding microenvironment. The findings support osteogenic differentiation, attachment, and remodeling [62].

Hydrogels provide an extracellular environment conducive to the growth of encapsulated cells and play an important role in regulating cell behavior. Mechanical and rheological properties of hydrogels can influence cell function, mechanotransduction, and cellular behaviors such as growth, migration, adhesion, self-renewal, differentiation, morphology, and differentiation [63,64]. Appropriate rheological properties are also essential for the development of hydrogels as bioink for printing tissue preparation of efficient drug delivery systems [63]. Moreover, an important aspect of 3D chondrogenic cell culture is that the scaffold allows the remodeling of its structure and rheological characteristics as cells differentiate along chondrogenic lineage. Some scaffolds have been reported to promote hUC-MSC adhesion, growth, and proliferation, as well as chondrogenic differentiation [64].

UC-based hydrogels are biocompatible for cells and exhibit support for cell attachment and proliferation. We made the same observations in our study, which showed that MSCs are compatible and proliferative in 10 mg/mL and 20 mg/mL concentrations in DUC hydrogels. Moreover, the present study explored the in vitro compatibility of MSCs with DUC hydrogel by determining the proliferation and chondrogenic differentiation of MSCs in vitro. When MSCs were uniformly cultured in the hydrogel, cellular condensations were observed in the scaffold. This phenomenon of mesenchymal cell condensation is very similar to the process observed in early chondrogenesis resulting from cell–cell and cell–matrix interactions [44]. Morphology of MSCs encapsulated in hydrogel scaffold appeared spindle-shaped. Cell viability assay showed a significant increase in proliferation of hydrogel-encapsulated cells from day 1 to day 7. A significant increase was observed in both the 10 mg/mL and 20 mg/mL hydrogels following 7 days of culture. The qualitative analysis further confirmed the viability of 3D-cultured MSCs, which resulted in progressive increase of the cell number with time, as the cells were viable and proliferative between days 1, 2, 3, 5, and 07 of culture in hydrogels. Previously, Liao et al. investigated the scaffold with an enhanced propagation capability and proliferation in in vitro culture, and in vivo vascularization [65]. The process of chondrogenic differentiation begins with the migration and homing of MSCs from the site of cartilage damage. TGF-β signaling is the initiation and a key player in the regulation of cartilage regeneration, which helps in the upregulation of extracellular matrix proteins involved in chondrogenesis, including fibronectin, N-cadherin, and Sox9. The transcription factor SOX-9 is an early marker for the development and maintenance of cartilage [66]; it supports cell survival and induces the expression of other chondrogenic genes such as *Aggrecan*, *TGFβ1*, *TGFβ2*, *BMP* [67,68]. TGF-βI also plays a role in inducing pre-cartilage condensation and retention of other members of the TGF-β family, including other matrix proteins such as collagen I, fibronectin, and tenascin [44]. Transcription factors have been shown to stimulate chondrocyte development in human umbilical cord MSCs (hUC-MSCs) [69]. The transcription factors SOX-9 and SIX-1 are known to stimulate chondrogenesis. When SOX-9 and SIX-1 were elevated, the transcription of BMP2, TGFβ1, SOX-9, and SIX-1 increased, effectively directing MSCs into chondrocytes. SIX-1 transcription is expected to be downregulated in differentiated cells. GDF5 is also an important marker of chondrogenesis. It expanded the number of prechondrogenic mesenchymal cell condensation and cartilaginous nodules without affecting the overall pattern of differentiation. GDF5 increases cartilage formation by promoting chondroprogenitor cell aggregation and amplifying the responses of cartilage differentiation markers [70] whereas aggrecan, a late chondrogenic marker, is elevated during chondrogenesis [71]. The adhesion and early transcription factors help in the aggregation of MSCs to secrete ECMs and induce differentiation, which leads to changes in their morphology. The cells then become hypertrophic and are subsequently replaced with newly formed hyaline cartilage cells [72]. Fibronectin present in the Wharton jelly matrix has been reported to aid in cartilage regeneration [73]. Lumican, a matrix protein involved in bone formation, has also been reported to be present in ECM of umbilical cord [74]. TGF-β, a major component of the ECM of Wharton jelly, is involved in the regulation of osteogenesis [75], and chondrogenic induction of MSCs by upregulating collagen-II [76]. It is reported that TGF-β plays a role in the synthesis of the ECM after chondrogenic differentiation [77,78]. The current study showed enhanced expression of key genes such as *TGF-*β1, *BMP2*, *SOX-9*, *SIX-1*, GDF-5, and *AGGRECAN* in stromal and chondro-inductive media after 28 days, which have previously been identified as crucial regulators of chondrogenesis [47]. However, when the data from the first group (2D culture MSCs/3D chondro-inductive hydrogel) and the second group (stromal media hydrogel/3D chondro-inductive hydrogel) were compared, it was clear that the first group had a higher expression of chondrogenic markers than the second group, while there was a significant statistical difference in MSCs’ chondrogenic differentiation pattern between 10 mg/mL and 20 mg/mL hydrogel.

## 5. Conclusions

It is concluded the DUC hydrogel functions as a 3D framework that provides the necessary microenvironment for proliferation and differentiation along the chondral lineage of WJ-MSCs in vitro. The hydrogel derived from human UC tissues is porous, bioactive, and biocompatible, which is capable of supporting MSC adhesion, proliferation, and migration by providing a tissue-like microenvironment. Moreover, it can effectively promote chondral tissue engineering in vitro. Such tissue-derived scaffolds have a variety of applications in 3D tissue models that facilitate cell–matrix interactions, and the fabrication of implantable scaffolding materials for guided tissue formation in vitro.

## Figures and Tables

**Figure 1 bioengineering-09-00239-f001:**
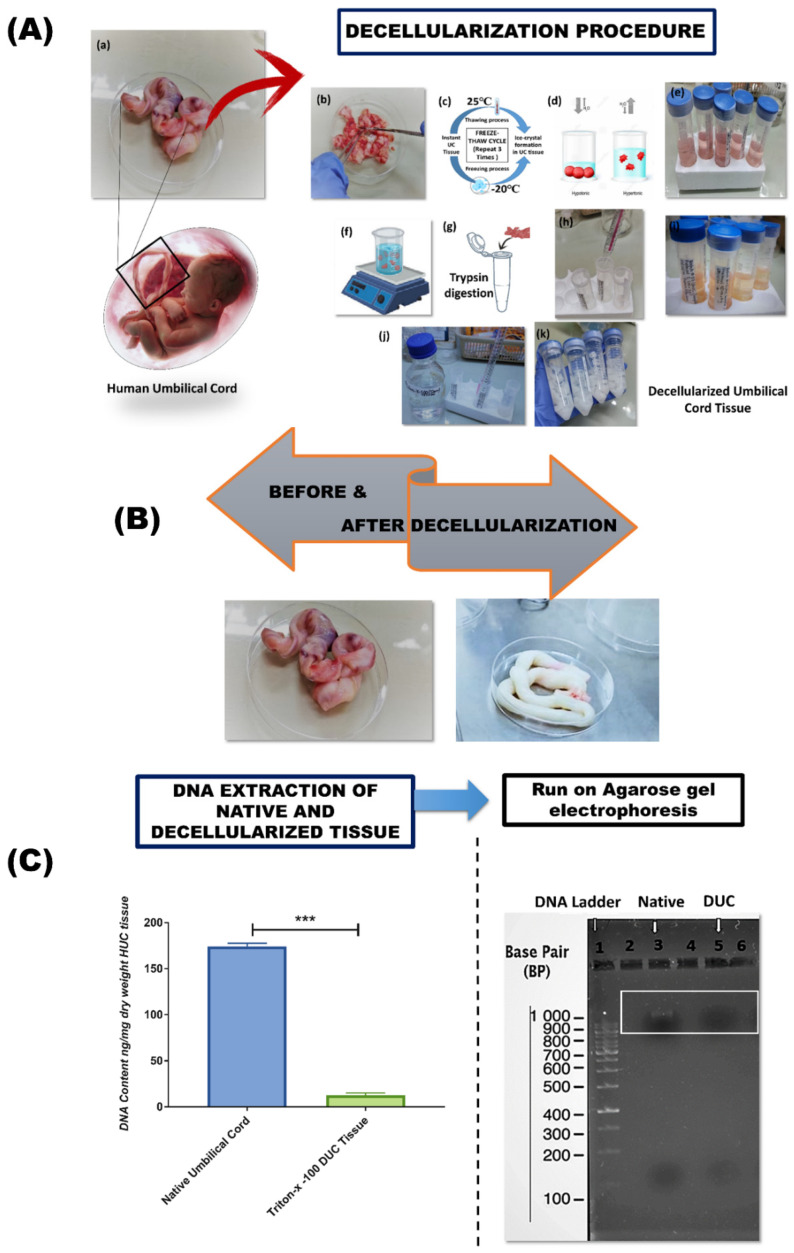
Diagrammatic representation of the procedure of decellularization of human umbilical cord tissue: (**A**) The human umbilical cord was collected (**a**), the cord sample was washed with deionized water to remove blood cells and minced in small pieces (about 6–7 cm) with a scalpel, and then washing was performed with water (**b**), tissue was frozen–thawed 3 times for 1 h each (**c**), the tissue was then subjected to osmotic shock using hypotonic and hypertonic solution (**d**,**e**), enzymatic digestion of tissue with 0.05% trypsin/EDTA for 2 h incubated at 37 °C (**f**–**i**), tissue sample was subjected to 1% Triton-x-100 (twice) incubation for at least 24 h. White transparent and highly hydrated decellularized umbilical cord tissue was obtained at the end of the process (**j**,**k**). (**B**) Human umbilical cord tissue before and after decellularization. (**C**) Quantitative and qualitative biochemical assessment of residual DNA in native and decellularized umbilical cord tissue. DNA quantification was performed before and after decellularization, which showed significantly less DNA in decellularized tissue compared to the native tissue (*n* = 3, *** *p* < 0.001).

**Figure 2 bioengineering-09-00239-f002:**
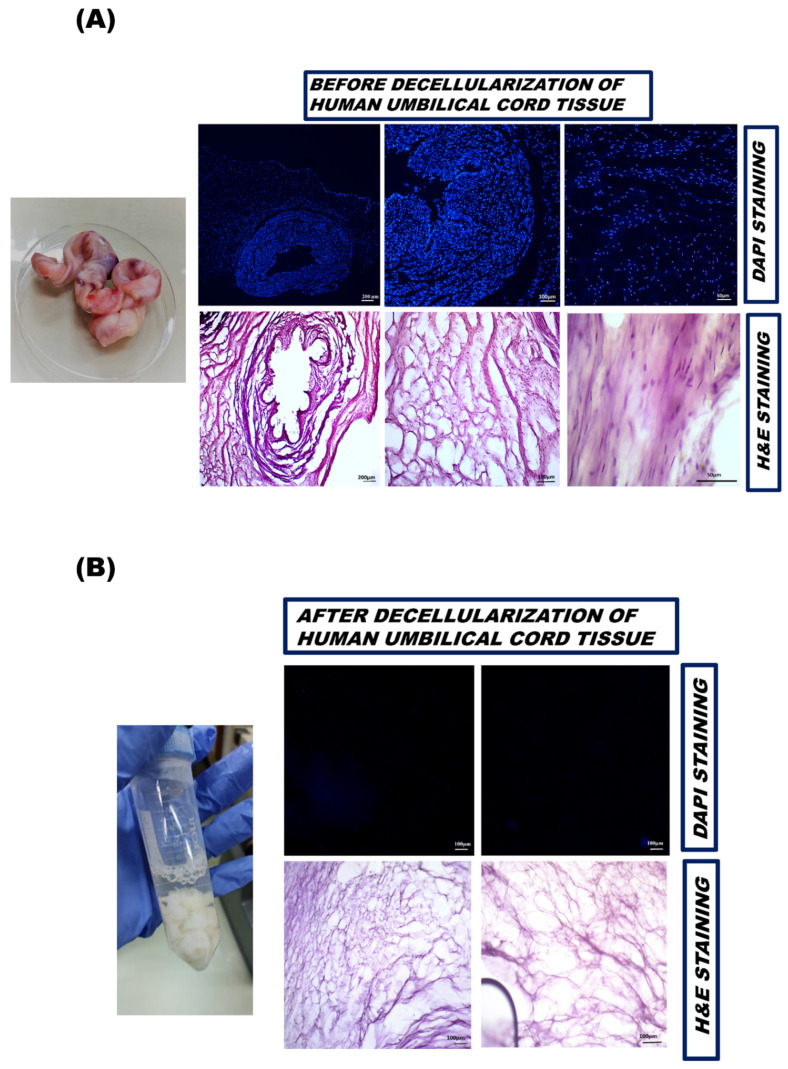
Histological analysis of nuclear content before and after decellularization of human umbilical cord tissue. (**A**) Histological analysis of nuclei in the native tissue was assessed by (DAPI) and (H&E) staining, (**B**) while no nuclei were observed in DUC tissue after decellularization.

**Figure 3 bioengineering-09-00239-f003:**
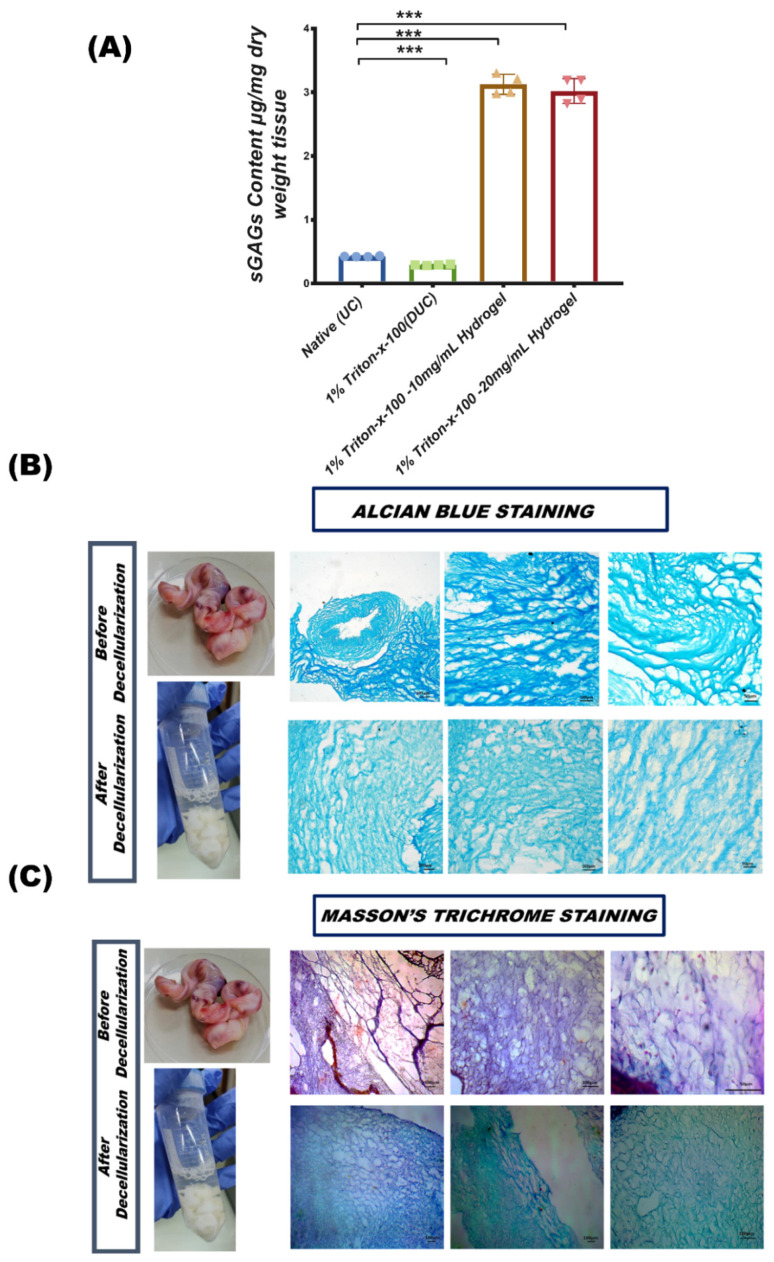
Quantitative and qualitative analysis of extracellular matrix. (**A**) sGAGs quantification by DMMB assay. (**B**) Alcian blue staining revealed the retention of sGAGs in the ECM of DUC (*n* = 3, per study group, *** *p* < 0.001): GAGs are light blue. (**C**) Masson’s trichrome staining showed collagen fiber images of native and DUC tissue.

**Figure 4 bioengineering-09-00239-f004:**
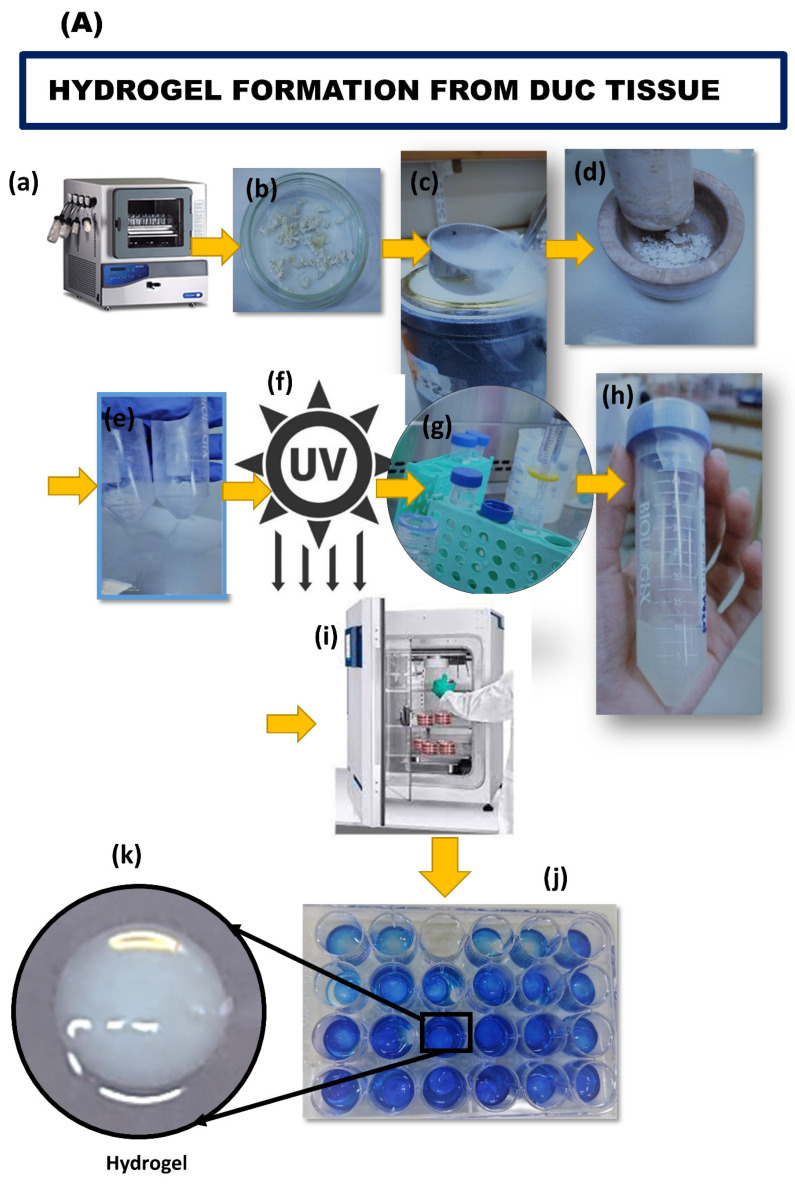
Schematic representation of decellularized tissue hydrogel (3D scaffold) and its biochemical and structural analysis: (**A**) Decellularized samples were lyophilized in a freeze dryer (**a**). The lyophilized tissues were then ground using mortar and pestle until they completely turned into a fine powder (**b**–**e**). Lyophilized powder was exposed to UV light for 30 min inside the biosafety cabinet (**f**). Pepsin/HCl digestion of lyophilized tissue powder (**g**) was carried out for at least 12 h to allow decellularized tissue to digest in this dissolution medium and to turn into pre-gel (**h**). Pre-gel was neutralized and then allowed to gel by incubating at 37 °C for 20–30 min to completely turn it into hydrogel (**i**–**k**). (**B**) FTIR spectra of DUC tissues, 10 mg, 20 mg pre-gel, and hydrogel reveal sharp peaks representing characteristic amide A, B, I, II, III, and GAGs regions are indicated by the dotted lines in the range of 4000–400 cm^−1^. (**C**) SEM micrograph of DUC tissue hydrogel revealed fibers rich in morphological characteristics with porosity, the pore size was significantly lower * *p* < 0.05 in 20 mg/mL hydrogel. (**D**) Percentage swelling behavior of 10 mg/mL and 20 mg/mL DUC hydrogel plot at various time points; 12 h, 24 h, 48 h, 56 h, 72 h, and 96 h.

**Figure 5 bioengineering-09-00239-f005:**
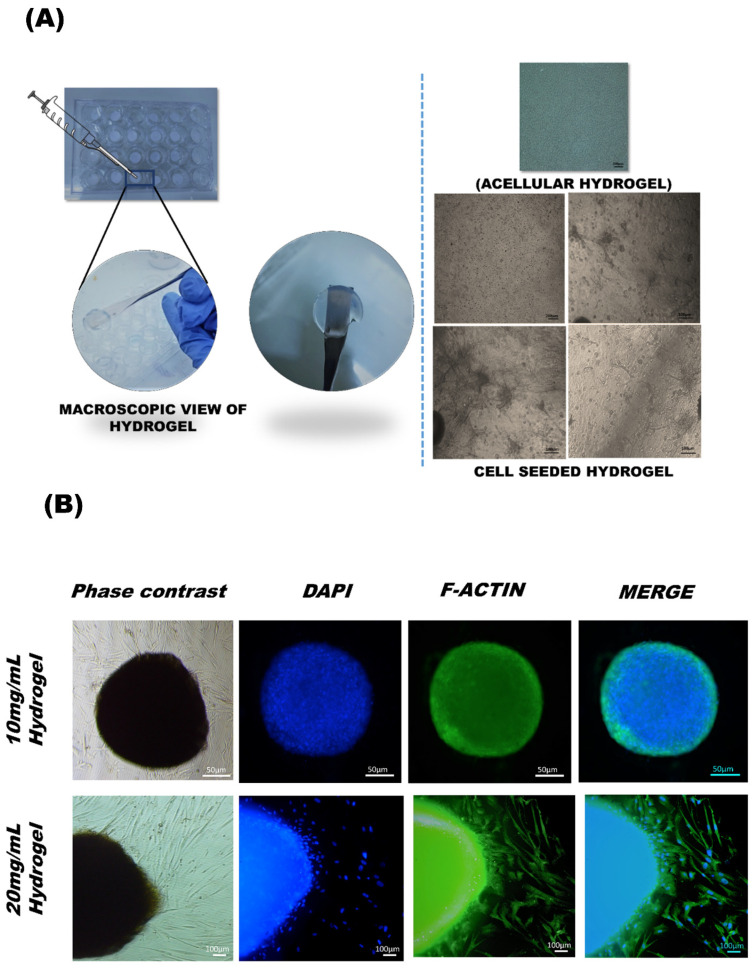
(**A**) Phase contrast images of hydrogel without and with cells show morphology of 3D encapsulated MSCs with a density of 0.2 million cells/mL. The cells in the decellularized hydrogel attached and maintained their 3D shape spindle shape after 24 h. (**B**) Immunofluorescence showing phase contrast, DAPI, and F-actin stained images of cell-seeded hydrogel.

**Figure 6 bioengineering-09-00239-f006:**
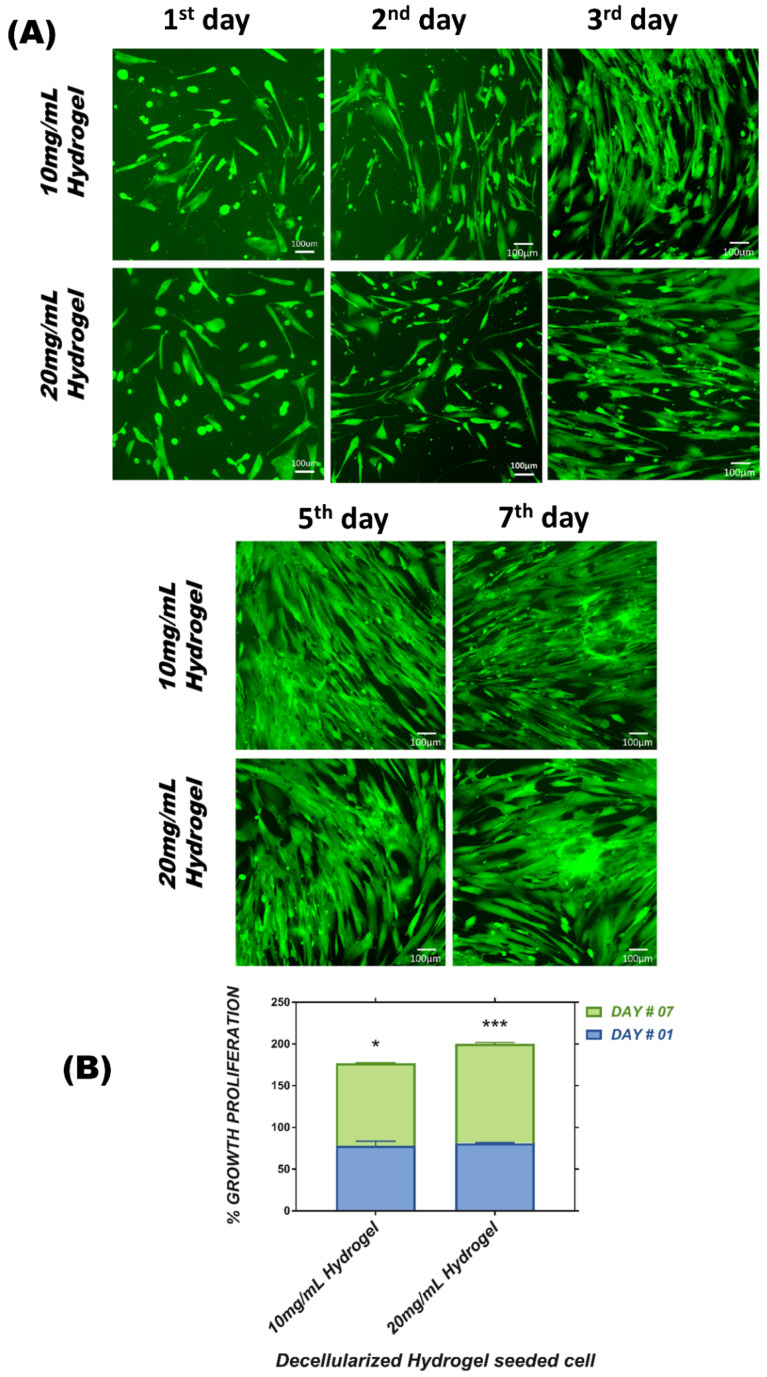
Proliferation and viability of cell-seeded hydrogel by Calcein-AM staining and Alamar blue assay: (**A**) Fluorescent microscopic images of DUC hydrogel, showing proliferation/ viability of cell-encapsulated hydrogel by Calcein-AM dye, at different time points, i.e., day 1, day 2, day 3, day 5, and day 7. (**B**) Alamar blue assay shows the proliferation of MSCs determined by fluorescence intensity on day 1 and day 7. Data are expressed as mean (*n* = 3) ± SD. The bar graph representing 1% Triton-X-100-treated 10 mg hydrogel with * *p* < 0.05 significance, 1% Triton-X-100-treated 20 mg hydrogel with *** *p* < 0.001 significance compared to day 1 of the respective hydrogel.

**Figure 7 bioengineering-09-00239-f007:**
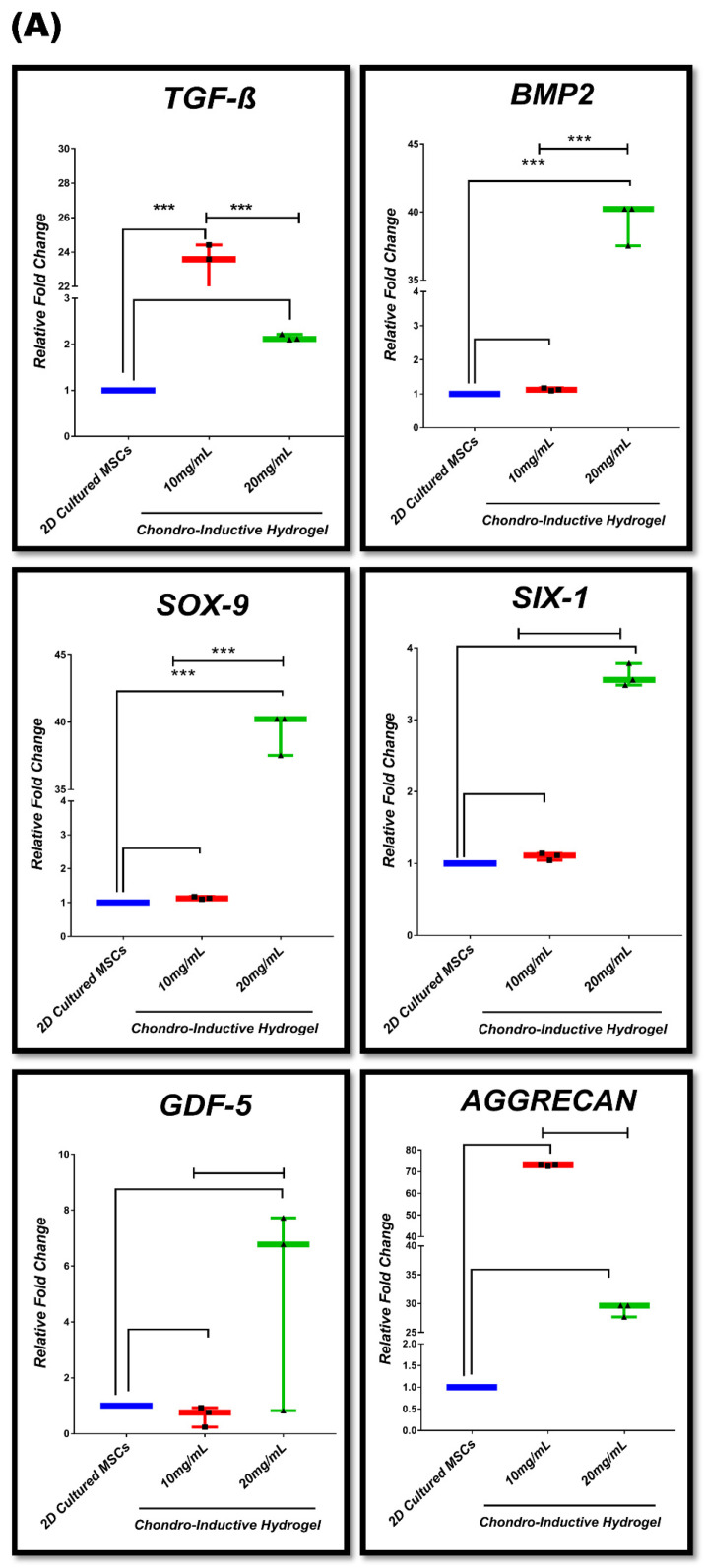
Gene expression profile of chondrogenic markers in 3D hydrogel-cultured MSCs: (**A**) Bar graphs representing quantitative 2-fold (2^−ΔΔCT^) changes in chondrogenic genes TGF-β, *BMP2*, *SOX-9*, *SIX-1*, *GDF-5*, and *AGGRECAN* in chondro-inductive hydrogel compared to 2D-cultured MSCs. (**B**) Expression of chondrogenic markers *TGF-Β*, *BMP2*, *SOX-9*, *SIX-1*, *GDF-5*, and *AGGRECAN* in stromal media 3D hydrogels compared to 2D-cultured MSCs as quantified by qPCR. (**C**) Chondrogenic genes in chondro-inductive media hydrogel compared to control hydrogel. Data were subjected to ANOVA with Bonferroni post hoc comparison and independent *t*-test. Significant increase of gene expression with * *p* < 0.05, ** *p* ≤ 0.01, *** *p* ≤ 0.001 was observed. Genes were normalized to *GAPDH*, and data represent the standard deviation of the mean of the three independent measurements.

**Table 1 bioengineering-09-00239-t001:** Primer sequences and annealing temperatures.

Gene Primers	Primer Sequences (5′-3′)	Annealing Temperatures (°C)
** *GAPDH* **	5′-CACCATGGGGAAGGTGAAGG -3′ 5′-AGCATCGCCCCACTTGATTT -3′	*58*
** *TGF-β1* **	5′-CAAGGCACAGGGGACCAG -3′ 5′-CAGGTTCCTGGTGGGCAG -3′	*58*
** *BMP2* **	5′-AGCTGGGCCGCAGGA -3′ 5′-TCGGCTGGCTGCCCT -3′	*58*
** *SOX-9* **	5′-CGGGCAAGGCTGACCTG -3′ 5′-GGTGCTGCTGATGCCGT -3′	*58*
** *SIX-1* **	5′-CTCCAGTCTGGTGGACTTGG-3′ 5′-AGCTTGAGATCGCTGTTGGT -3′	*58*
** *GDF5* **	5′-CACATCCCAAGAGCCCCTTC -3′ 5′-GCCCAGGTGAGGAGAAATGG -3′	*58*
** *ACCAN* **	5′- AGTTCTGTGAATCTCACAATGCC-3′ 5′- CCAGAGGGACTGACATTTTCTTG-3′	*58*

**Table 2 bioengineering-09-00239-t002:** Experimental groups for gene expression analysis of DUC hydrogel (10 mg/mL or 20 mg/mL).

**Group #01**	2D-cultured MSCs in stromal media (control) vs. 10 mg/mL chondro-inductive hydrogel 20 mg/mL chondro-inductive hydrogel
**Group #02**	2D-cultured MSCs in stromal media (control) vs. 10 mg/mL stromal media hydrogel 20 mg/mL stromal media hydrogel
**Group #03**	10 mg/mL stromal media hydrogel (control) 20 mg/mL stromal media hydrogel vs. 10 mg/mL chondro-inductive hydrogel 20 mg/mL chondro-inductive hydrogel

## Data Availability

Not applicable.

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
