# Peer review of "Decellularized Human Umbilical Tissue-Derived Hydrogels Promote Proliferation and Chondrogenic Differentiation of Mesenchymal Stem Cells"

_bioengineering, 2022, doi:10.3390/bioengineering9060239_

Round 1
Author Response
We appreciate the time and effort of the reviewers for providing valuable feedback on the manuscript. We are grateful to the reviewers for their insightful comments. We have highlighted the changes within the manuscript. Here is a point-by-point response to the reviewers’ comments and concerns.
In Figure 1C, the author stated that DNA isolated from decellularized samples did not show any visible bands, which confirms the elimination of DNA in DUC samples. But in Figure 1C, the bands appear with less intensity. So is the author’s statement correct?
We have corrected the sentence in the revised version. The DNA content in our study decreased from 174.15ng ± 2.45 to 12.63ng ± 2.42 of DNA/mg dry weight of tissue after Decellularization treatment. The data from DAPI and H&E staining were also confirming the removal of intact DNA. It is reported that DNA amount of 50 ng/mg of tissue is acceptable for in vivo transplantation and will not elicit host (Crapo et al., 2011; Penolazzi et al., 2020; Safari, et al., 2019; Faulk, et al., 2014).
- Crapo, P. M., Gilbert, T. W., and Badylak, S. F. (2011). An overview of tissue and whole organ decellularization processes. Biomaterials 32, 3233–3243. doi: 10.1016/j.biomaterials.2011.01.057
- Penolazzi, L., Pozzobon, M., Bergamin, L. S., D’Agostino, S., Francescato, R., Bonaccorsi, G., ... & Piva, R. (2020). Extracellular matrix from decellularized Wharton’s jelly improves the behaviour of cells from degenerated intervertebral disc. Frontiers in bioengineering and biotechnology, 8, 262.
- Safari, F., Fani, N., Eglin, D., Alini, M., Stoddart, M. J., & Baghaban Eslaminejad, M. (2019). Human umbilical cord‐derived scaffolds for cartilage tissue engineering. Journal of Biomedical Materials Research Part A, 107(8), 1793-1802.
- Faulk, D. M., Johnson, S. A., & Badylak, S. F. (2014). Decellularized biological scaffolds for cardiac repair and regeneration. In Cardiac regeneration and repair(pp. 180-200). Woodhead Publishing.
The same was observed in Figure 2B, where some DAPI staining was observed at 200 µm. So, how does this further confirm that complete removal of DNA has not been observed? Will the small amount of DNA material present in the tissue samples impact the other analysis?
Figure 2B, show DAPI staining for decellularized tissue. The light color is from the background of decellularized tissue, there are no intact nuclei, and was further confirmed with H and E staining. Therefore, the statement is correct that DNA is eliminated from decellularized tissue.
The authors have maintained images with same scale bars in Figures 2A and 2B.
The scale bar is corrected in the revised version.
The authors had found the ECM Component Quantification. How about the bicomptabilty of developed hydrogels?
To check the biocompatibility of hydrogel Alamar blue assay, cell proliferation assay, live-cell imaging assay, F-actin staining of cells seeded hydrogel, and cell differentiation assay were performed. Alternatively, literature support is provided.
Improve the language, and if possible, find a native speaker or a company to polish the language.
The language of the revised manuscript is improved.
Too many typical and syntax errors are observed, so you need to improve those things. For example, in line 344, the author has written mg, mL. Make it correct as mg/mL. In line 315, two words are clubbed together (whichshows). Make it correct. Like this, so many typical errors are there.
We thank the reviewer for pointing this out. These corrections are made throughout the manuscript in the revised version.

Reviewer 2 Report
This manuscript investigates the preparation of hydrogels derived from decellularized human umbilical tissue and their effect on the proliferation and differentiation of MSCs. However, the advantage and disadvantage of this new material are unclear comparing with other researches previously reported. This point should be discussed clearly. Moreover, there are no data of rheology or porous structure of prepared hydrogels. The data are essential to design scaffolds and should be added.
Discussion of results is quite poor. Taken together, I cannot recommend this manuscript to deserve publication in this style.
- There is no clear explanation in the Materials Method section what do the words ‘’10 mg/mL’’ and ‘’20 mg/mL’’ stand for? The explanations should be added.
- In the Results section of ‘’Decellularization of Human Umbilical Cord Tissue’’, there is a sentence ‘’The mentioned decellularization protocol displayed promising results for the removal of cells, while maintaining the three-dimensional structure as well as the composition of the ECM.’’. However, there are no data in this section (3.1) deserve to discuss about the maintaining of the three-dimensional structure during decellularization. This sentence should be deleted or revised.
- The rheological property of hydrogels must depend on the procedure of decellularization and the type of human resource. This point should be discussed or described clearly.
- How about the degradability of hydrogels prepared? This must affect the conditions and fate of MSCs differentiation. This point should be described by adding some data or quoting some related literatures.
- In lines 344 to 346, the unit of hydrogels concentration, mg/mL, is not unified. This should be carefully corrected.
- In the Results section of ‘’Glycosaminoglycans (GAGs) Content of Extracellular Matrix’’, the intensity of stained GAG is described. Does the diameter of stained fiber become small through the decellularization process? This point should be described clearly. I think it is possible that the GAG is denatured through the decellularization process.
- The rheology and porous structure of hydrogels greatly affect the attachment, proliferation and differentiation of cells seeded. The data should be added.
- The qPCR is conducted evaluate the chondrogenic differentiation of MSCs, and the results are reasonable. How about the change of scaffolds’ rheology during the cell culture? Generally, it is scientifically difficult to evaluate the chondrogenic differentiation of MSCs. I think it is necessary to evaluate it by other methods than qPCR. There are some researches to discuss the chondrogenic differentiation of MSCs in hydrogels with scaffolds’ rheology.
Author Response
We appreciate the time and effort of the reviewers for providing valuable feedback on the manuscript. We are grateful to the reviewers for their insightful comments. We have highlighted the changes within the manuscript. Here is a point-by-point response to the reviewers’ comments and concerns.
This manuscript investigates the preparation of hydrogels derived from decellularized human umbilical tissue and their effect on the proliferation and differentiation of MSCs. However, the advantage and disadvantage of this new material are unclear comparing with other researches previously reported This point should be discussed clearly.
We agree with the reviewer, we have added points regarding the advantage and disadvantages of this new material in the introduction as well as in the discussion section. In a nutshell, the novel hydrogel, made from DUC tissue has more advantageous because it is rich in proteoglycans, hyaluronic acid, elastin fiber, collagen, and growth factors required for proliferation, matrix synthesis, and promotes effective tissue remodelling. Collagen is the main component of wharton’s jelly constituting 50% by weight. Hyaluronic acid is the most abundant component of Wharton’s jelly Glycosaminoglycan (GAGs). ECM extracted from human umbilical cord tissue has some important rheological features including high content of GAGs, short gelation time, and a high elastic modulus which make it superior to other tissue derived and synthetic scaffolds (Kurtz A, Oh SJ 2012; Badylak SF 2014; Freytes et al., 2008). Furthermore, it showed decreased risk of inflammation, and host immune reactions due to the removal of antigenic cellular components of the tissue which is normally associated with other kinds of scaffold. It is considered as medical waste and is therefore relatively easily accessible human tissue while the maintenance of sterility is one of the drawbacks related to DUC hydrogel. Taken together all these facts, UC could be a better source for making hydrogels for tissue regeneration.
- Kurtz A, Oh SJ (2012) Age related changes of the extracellular matrix and stem cell maintenance. Prev Med (Baltim)
- Badylak SF (2014) Decellularized allogeneic and xenogeneic tissue as a bioscaffold for regenerative medicine: Factors that influence the host response. Ann Biomed Eng
- Freytes DO, Martin J, Velankar SS, et al. (2008) Preparation and rheological characterization of a gel form of the porcine urinary bladder matrix. Biomaterials
Moreover, there are no data of rheology or porous structure of prepared hydrogels. The data are essential to design scaffolds and should be added.
Thank you for this suggestion. It would have been interesting to explore this aspect. We have added the suggested content in our manuscript using various tests, rheological properties of the hydrogels such as gelation time, porosity, storage, and loss modulus all of which contribute towards evaluating the given hydrogel for the intended application. Rheology provides the structure-property relationships to understand the various mechanisms of hydrogels which can tune the cross-linking density to assess the mechanical strength. In order to characterize hydrogel via different perspective, we have conducted FTIR spectroscopy to check its gelation ability and biochemical composition, swelling behaviour was analyzed to check porosity and swelling behavior effect, and topographical analysis via SEM to study its appearance and pore size diameter which is one the key aspect of maintaining cellular morphology and proliferation. All these data have been added as per your instruction.
Discussion of results is quite poor. Taken together, I cannot recommend this manuscript to deserve publication in this style.
Discussion section is improved as per recommendation; we have added further data to strengthen our study and cited relevant literature. We have made the change in writing style to make it better version for publication.
There is no clear explanation in the Materials Method section what do the words ‘’10 mg/mL’’ and ‘’20 mg/mL’’ stand for? The explanations should be added.
The methods section is revised and explanation is added for gel concentrations, 10 and 20 mg/mL hydrogel is made by taking 10 and 20 mg powder of DUC tissue and digested with 1mLpepsin/ HCl solution. Moreover, we used two concentrations of hydrogel in order to check the conc. dependent effect on cells compatibility as well as on differentiation
- In the Results section of ‘’Decellularization of Human Umbilical Cord Tissue’’, there is a sentence ‘’The mentioned decellularization protocol displayed promising results for the removal of cells, while maintaining the three-dimensional structure as well as the composition of the ECM.’’. However, there are no data in this section (3.1) deserve to discuss about the maintaining of the three-dimensional structure during decellularization. This sentence should be deleted or revised.
The mentioned sentence is deleted from the revised version, as suggested by reviewer.
- The rheological property of hydrogels must depend on the procedure of decellularization and the type of human resource. This point should be discussed or described clearly.
We have included an additional figure (4) in the revised manuscript, the include FTIR analysis of the decellularized native tissue, pre-gel, and hydrogel. SEM data is also included in the revised version, which highlights the topography of the gel and pore size. We also performed the swelling test to measure the water content in the gel. We have provided literature related to UC based hydrogel rheology in the discussion section.
- How about the degradability of hydrogels prepared? This must affect the conditions and fate of MSCs differentiation. This point should be described by adding some data or quoting some related literatures.
We have provided the literature support in the revised version. Biodegradation of hydrogels is essential to generate suitable room for the arriving occupants, in addition to cell proliferation and microstructural remodelling. Regardless of the natural or synthetic polymers employed in their production, the breakdown of hydrogels is primarily influenced by temperature, pH, UV irradiation, ultrasound, and enzymes, among other factors. Previous study reported the degradation of ECM proteins in decellularized adipose tissue (DAT) hydrogel by MMPs allows the cells to remodel the surrounding microenvironment. The findings support osteogenic differentiation, attachment, and remodelling of hydrogel (Mohiuddin, et al. 2020). Moreover, the hydrogel fabricated from human umbilical cord tissue has similar fabrication method and previous studies reported the stable nature of hydrogel. Further, we will analyze this aspect in vivo studies in future.
- Mohiuddin OA, Motherwell JM, Rogers E, et al. (2020) Characterization and Proteomic Analysis of Decellularized Adipose Tissue Hydrogels Derived from Lean and Overweight/Obese Human Donors. Adv Biosyst
- In lines 344 to 346, the unit of hydrogels concentration, mg/mL, is not unified. This should be carefully corrected.
We have corrected and Explanation is added as well for gel concentrations.
- In the Results section of ‘’Glycosaminoglycans (GAGs) Content of Extracellular Matrix’’, the intensity of stained GAG is described. Does the diameter of stained fiber become small through the decellularization process? This point should be described clearly. I think it is possible that the GAG is denatured through the decellularization process.
The GAG content in the hydrogel was quantified and presented in figure 3A. Alcian blue stain showed sulfated glycosaminoglycans (GAGs) in the ECM of tissue, producing an intense blue colour around the cells matrix. The results were found to be positive for native and decellularized tissue but the intensity was slightly different in DUC, because of the denaturation of GAGs. It’s a hydrophilic molecule that would be affected by Triton X-100 treatment which disrupt DNA-protein, lipid-lipid, and lipid-protein interactions while maintaining native protein structures. the GAGs are associated with collagen fibers that’s why the images showed the reduction of stained fiber diameter after decellularization procedure that affect the entire proteoglycans and this was observed previously in several decellularization studies (Jadalannagari, et al., 2017; Kim, et al., 2021, November) and the same trend was observed in the current studies. But the detergent we used in our study is producing less harsh environment then Ionic detergents, e.g. sodium dodecyl sulfate (SDS) which completely solubilize cell and nucleic membranes and denature protein as reported in literature.
Jadalannagari, S., Converse, G., McFall, C., Buse, E., Filla, M., Villar, M. T., ... & Aljitawi, O. S. (2017). Decellularized Wharton’s Jelly from human umbilical cord as a novel 3D scaffolding material for tissue engineering applications. PLoS One, 12(2), e0172098.
Kim, M. K., Jeong, W., & Kang, H. W. (2021, November). Effect of detergent type on the performance of liver decellularized extracellular matrix-based bio-inks. 한국바이오칩학회.
- The rheology and porous structure of hydrogels greatly affect the attachment, proliferation and differentiation of cells seeded. The data should be added.
The data is added in the revised manuscript. The pore size and swelling index of hydrogels has been determined and data has been added in the results to enhance the physical characterization of the hydrogels. We realize that the determination of rheological properties is an important part of hydrogel characterization, however, we performed several rheological tests right now. We have added a paragraph in the discussion as well to highlight the significance of rheological characterization of hydrogels and that it needs to be done in subsequent studies to strengthen the present findings.
- The qPCR is conducted evaluate the chondrogenic differentiation of MSCs, and the results are reasonable. How about the change of scaffolds’ rheology during the cell culture? Generally, it is scientifically difficult to evaluate the chondrogenic differentiation of MSCs. I think it is necessary to evaluate it by other methods than qPCR. There are some researches to discuss the chondrogenic differentiation of MSCs in hydrogels with scaffolds’ rheology.
We have evaluated the chondrogenic differentiation by qPCR analysis. We have supported our finding with literature and it confirm that cells in the hydrogel are differentiated into chondrogenic lineage. We agree with the reviewer that further elaborating scaffolds’ rheology. In future we intended to evaluate rheological profile along chondrogenic differentiation. We agree that this is a potential limitation of our study. We have added relevant literature as per your recommendation. However, we plan for future experiment to conduct chondrogenic differentiation of MSCs in hydrogels with scaffolds’ rheology.

Reviewer 3 Report
1.
Please provide full name when firstly using an abbreviation. For example, GAGs, IGF, PDGF from the introduction. Please check though the full text.
2.
Line 133 section 2.4. Please provide the tissue sectioning procedure, and the procedure before staining regarding OCT or paraffin embedding.
3.
Line 188-221. In this study, the MSC culture density in the hydrogel was initially 0.2 M / 200 ul in section 2.7 and the same density was used in section 2.10 for gene expression. However, the density for Alamar blue assay became 1200 / 100 ul in section 2.9.2. In figure 4, the caption showed 0.2 M / ml density for the 3D cell encapsulation for phase contrast images. Please provide a reasonable explanation of using different concentrations of MSCs.
4.
Line 240, table 2. The sequence of reversed primer for ACCAN seems not belonging to the homo ACCAN mRNA. Please repeat the PCR with a valid primer pair. If the authors used the correct one already, please fix the sequence in the table.
5.
In this study, the authors claimed, “we first characterized decellularized human UC scaffold and then tested its chondro-inductive capability.” from line 96. However, the most characteristic works were done for the DM, which is not the final UC-based hydrogel scaffold that the authors cultured with MSCs to present its chondro-inductivity. The structure and properties of the scaffold changed dramatically after gelation. The only characterization in this paper for the final scaffold is phase contrast microscopy. Please consider to provide more characterization on the scaffold itself, such as hydrogel swelling test. SEM, mechanical test.
6.
The authors stated in Discussion, line 520 that TDS is biomimetic and suitable for chondrogenic application. It is true that the scaffold in this study contains GAGs, collagen, and hyaluronic acid that would promote the proliferation and differentiation of MSCs. However, the microstructure will be changed due to lyophilization, pepsin treatment, and gelation. The authors should discuss more about the final hydrogel scaffold that encapsulated cells, showing how the hydrogel can “recapitulate the key physicochemical characteristics of native human tissue”.
7.
The authors showed 5 groups in table 1. However, the naming was changed in later PCR section 3.6, line 426. Does “chondro-inductive hydrogel” mean “chondro-inductive media 3D Hydrogel”? Please make sure the naming is consistent. In the PCR study, the absence of 2D culture with chondrogenic differentiation media weakens the significance of the study. Furthermore, please add a paragraph in the Discussion section to discuss the PCR result to better support the chondrogenic statements. The meaning of the expression of each picked marker gene should be discussed. The comparison between 2D and 3D culture, and between the effects of chondrogenic media on 10 mg/ml and 20 mg/ml hydrogels should be discussed.

Author Response
This study showed a novel method to fabricate a biomimetic hydrogel scaffold for cartilage tissue regeneration. Primer human MSCs were 3D encapsulated within the hydrogel in this study and showed the biocompatibility and chondro-inductivity of the scaffold. There are some experiments suggested to further strengthen the paper. There are also some confusing statements need to be clarified or fixed.
We appreciate the time and effort of the reviewers for providing valuable feedback on the manuscript. We are grateful to the reviewers for their insightful comments. We have highlighted the changes within the manuscript. Here is a point-by-point response to the reviewers’ comments and concerns.
- Please provide full name when firstly using an abbreviation. For example, GAGs, IGF, PDGF from the introduction. Please check though the full text.
The full form of all the abbreviations is provided in the revised manuscript.
- Line 133 section 2.4. Please provide the tissue sectioning procedure, and the procedure before staining regarding OCT or paraffin embedding.
This section is added in the revised manuscript as per your instruction cited with references.
- Line 188-221. In this study, the MSC culture density in the hydrogel was initially 0.2 M / 200 ul in section 2.7 and the same density was used in section 2.10 for gene expression. However, the density for Alamar blue assay became 1200 / 100 ul in section 2.9.2. In figure 4, the caption showed 0.2 M / ml density for the 3D cell encapsulation for phase contrast images. Please provide a reasonable explanation of using different concentrations of MSCs.
Lower cell seeding density is required for cell proliferation assay. As the cells are growing and proliferating, therefore, a free surface/space is required for them. We need to observe the cells over a period of time and need to track them if they are proliferating, therefore low cell seeding density is required. While for analysis of cell differentiation higher cell density is required, as we aim to differentiate the cells.
- Line 240, table 2. The sequence of reversed primer for ACCAN seems not belonging to the homo ACCAN mRNA. Please repeat the PCR with a valid primer pair. If the authors used the correct one already, please fix the sequence in the table.
We have checked the primer sequence for aggrecan and we have corrected the gene sequence in the revised version of manuscript.
- In this study, the authors claimed, “we first characterized decellularized human UC scaffold and then tested its chondro-inductive capability.” from line 96. However, the most characteristic works were done for the DM, which is not the final UC-based hydrogel scaffold that the authors cultured with MSCs to present its chondro-inductivity. The structure and properties of the scaffold changed dramatically after gelation. The only characterization in this paper for the final scaffold is phase contrast microscopy. Please consider to provide more characterization on the scaffold itself, such as hydrogel swelling test. SEM, mechanical test.
We have included additional data in the revised manuscript according to your recommendation. We have added FTIR analysis of the decellularized native tissue, pre-gel, and hydrogel to confirm the gelation procedure from tissue to hydrogel. SEM data is also included in the revised version, which highlight the topography of the gel and pore size. We also performed the swelling test to measure the water content and imbibition capacity in the gel. Porosity of hydrogel was also analyzed.
- The authors stated in Discussion, line 520 that TDS is biomimetic and suitable for chondrogenic application. It is true that the scaffold in this study contains GAGs, collagen, and hyaluronic acid that would promote the proliferation and differentiation of MSCs. However, the microstructure will be changed due to lyophilization, pepsin treatment, and gelation. The authors should discuss more about the final hydrogel scaffold that encapsulated cells, showing how the hydrogel can “recapitulate the key physicochemical characteristics of native human tissue”.
The cross linking was characterized by comparative analysis of FTIR spectra of the lyophilized sample which is evaluated throughout the procedure from enzymatic digestion to gel preparation and it confirms the similar biochemical composition only the intensity of cross link was increased from tissue to hydrogel as you can see the figure 5A. Proliferation and viability assay was also performed and other characterization of scaffold which support its physiochemical properties. These highlighted points are discussed in detail in introduction and discussion section.
- The authors showed 5 groups in table 1. However, the naming was changed in later PCR section 3.6, line 426. Does “chondro-inductive hydrogel” mean “chondro-inductive media 3D Hydrogel”? Please make sure the naming is consistent. In the PCR study, the absence of 2D culture with chondrogenic differentiation media weakens the significance of the study. Furthermore, please add a paragraph in the Discussion section to discuss the PCR result to better support the chondrogenic statements. The meaning of the expression of each picked marker gene should be discussed. The comparison between 2D and 3D culture, and between the effects of chondrogenic media on 10 mg/ml and 20 mg/ml hydrogels should be discussed.
The naming is corrected now and was made consistent. Yes, the mentioned group is chondro-inductive media 3D hydrogel. We added a paragraph in the discussion and highlighted the importance of the gene for chondrogenic differentiation. We compared the gene expression between 2D and 3D in chondro-inductive media, stromal media, and 3D stromal and chondro-inductive media.

Round 2
Reviewer 2 Report
The content has been carefully and seriously revised by adding sentences and explanation based on the comments. Based on this, I recommend the revised manusript to deserve the publication.